# Dialysate calcium, alfacalcidol, and clinical outcomes: A post-hoc analysis of the J-DAVID trial

**Kunitoshi Iseki**[1]*, **Daijiro Kabata**[2], **Tetsuo Shoji**[3,4], **Masaaki Inaba**[5], **Masanori Emoto**[4,5,6], **Katsuhito Mori**[6], **Tomoaki Morioka**[5], **Shinya Nakatani**[5], **Ayumi Shintani**[2]

**1** Nakamura Clinic, Urasoe, Okinawa, Japan, **2** Department of Medical Statistics, Osaka City University Graduate School of Medicine, Osaka, Japan, **3** Department of Vascular Medicine, Osaka City University Graduate School of Medicine, Osaka, Japan, **4** Vascular Science Center for Translational Research, Osaka City University Graduate School of Medicine, Osaka, Japan, **5** Department of Metabolism, Endocrinology and Molecular Medicine, Osaka City University Graduate School of Medicine, Osaka, Japan, **6** Department of Nephrology, Osaka City University Graduate School of Medicine, Osaka, Japan

* chihokun_ohra@yahoo.co.jp

## Abstract

The selection of dialysate calcium concentration (D-Ca) is still controversial among chronic hemodialysis (HD) regimens. We examined the trajectories of CKD MBD parameters among the J-DAVID trial participants to see the effect of D-Ca and alfacalcidol. The trial was an open-label randomized clinical trial including 976 HD patients with intact PTH of 180 pg/mL or lower which compared the users of vitamin D receptor activator (oral alfacalcidol) and non-users over a median of 4 years. The main D-Ca used at baseline were 3.0 mEq/L in 70% and 2.5 mEq/L in 25%, respectively. The primary endpoint was the composite of fatal and non-fatal cardiovascular events and the secondary endpoint was all-cause mortality. Multivariable Cox proportional hazard regression analyses in which D-Ca was included as a possible effect modifier and serum laboratory data as time-varying covariates showed no significant effect modification for composite cardiovascular events or all-cause mortality. This post hoc analysis showed that the effects of alfacalcidol on cardiovascular outcomes were not significantly modified by D-Ca.

## Introduction

The hemodialysis (HD) regimen for kidney failure remained experience-based. They consist of session time and frequency of HD, the material of membrane and size of the dialyzer, vascular access, and dialysate composition. Dialysate composition includes sodium, potassium, glucose, acetate, bicarbonate, and calcium. Among them, dialysate calcium (D-Ca) may play an important role in the management of chronic kidney disease-mineral and bone disorders (CKD-MBD). CKD-MBD is one of the major complications leading to cardiovascular disease (CVD), vascular calcification, bone fracture, and all-cause mortality [1].

Ideally, the dialysis regimen should be individualized [2] as the patient's status such as age, body size, and co-morbid conditions are variable. However, most dialysis units are using a

**Data Availability Statement:** Data cannot be shared publicaly because of the regulation of Research Committee of JDAVID (pricipal researcher is Dr Testuo Shoji). The original paper

was already published in the JAMA. The current paper is the posthoc analysis frim the JAMA paper.

**Funding:** Funding: This study was supported by the grant to TS from The Kidney Foundation, Japan. Author Contributions Conceptualization: Kunitoshi Iseki, Tetsuo Shoji Funding acquisition: Tetsuo Shoji Investigation: Kunitoshi Iseki, Daijiro Kabata Methodology: Kunitoshi Iseki, Daijiro Kabata Supervision: Tetsuo Shoji Visualization: Kunitoshi Iseki, Daijiro Kabata Writing-original draft: Kunitoshi Iseki Writing-review & edition: all co-authors.

**Competing interests:** Kunitoshi Iseki received personal fees from Kyowa Kirin, Bayer Yakuhin, Torii Pharmaceutical, Mochida Pharmaceutical, and Kissei Pharmaceutical; and received consultation fee from the MediGate Digital Health K.K., Daijiro Kabata received personal fees from Chugai Pharmaceutical; research grant and consultation fee for statistical analysis from Kyowa Kirin; and personal fees and a research grant from Bayer Yakuhin outside the submitted work. Ayumi Shintani received personal fees from Chugai Pharmaceutical, Bayer Yakuhin, and Ono Pharmaceutical; and personal fees and research grants from Kyowa Kirin, Takeda Pharmaceutical, and Daiichi Sankyo outside the submitted work. Shinya Nakatani received personal fees from Chugai Pharmaceutical, Kyowa Kirin, Bayer Yakuhin, Ono Pharmaceutical, Kissei Pharmaceutical, and Torii Pharmaceutical. Tomoaki Morioka received personal fees from Ono Pharmaceutical. Katsuhito Mori received personal fees from Ono Pharmaceutical; he served as the principal investigator of a clinical trial for Kyowa Kirin; and personal fees and a research grant from Mitsubishi Tanabe outside the submitted work. Masaaki Inaba received personal fees from Chugai Pharmaceutical, Kyowa Kirin, Bayer Yakuhin, Ono Pharmaceutical, Kissei Pharmaceutical, and Torii Pharmaceutical outside the submitted work. Masanori Emoto received personal fees from Kissei Pharmaceutical, AstraZeneca, and Torii Pharmaceutical; and personal fees and research grants from Chugai Pharmaceutical, Kyowa Kirin, Ono Pharmaceutical, Nippon Boehringer Ingelheim, and Mitsubishi Tanabe outside the submitted work. Tetsuo Shoji received personal fees from Chugai Pharmaceutical, Kyowa Kirin, and Kissei Pharmaceutical; and personal fees and research grants from Bayer Yakuhin, and Ono Pharmaceutical outside the submitted work. This does not alter our adherence to PLOS ONE policies on sharing data and materials.

central dialysis fluid delivery system (CDDS), at least in Japan [3]. Dialysate is selected and fixed in each facility according to the choice of the primary physician. Currently, D-Ca is mostly 2.5 to 3.0 mEq/L in Japan as suggested by KDIGO clinical practice guideline (CPG) based on observational studies [1].

Many observational studies showed that the use of vitamin D receptor activator (VDRA) was associated with a lower risk of all-cause mortality [4, 5], cardiovascular mortality [6], and incident cardiovascular events [7] in hemodialysis patients. In addition, a large cohort study [4] showed the association between VDRA use and all-cause mortality was found regardless of intact parathyroid hormone (intact PTH) levels. VDRAs are shown to have pleiotropic actions which are potentially beneficial in patients with chronic kidney disease [8]. However, in the Japan Dialysis Active Vitamin D (J-DAVID) trial [9], treatment with alfacalcidol did not reduce the risk of composite cardiovascular events or the risk of all-cause mortality in hemodialysis patients without secondary hyperparathyroidism. Then, we hypothesized that dialysate calcium concentration (D-Ca) might have influenced the serum parameters of CKD-MBD and modified the effects of alfacalcidol on these clinical outcomes in the trial. We tested this hypothesis in this post hoc analysis of the J-DAVID trial data.

## Methods

We used the data obtained from the participants of the J-DAVID trial [9, 10] to examine the effect of D-Ca on the CKD MBD parameters and outcomes. In brief, during the recruitment period from July 1, 2008, to January 26, 2011, a total of 1,289 patients were assessed for eligibility and 976 patients were randomized to VDRA (oral alfacalcidol) or no VDRA (control). The median follow-up was 4 years, and the all-cause mortality rates were 18.2% in VDRA and 16.8% in control, respectively. During the study period, serial data of mineral metabolism were collected such as corrected Ca, phosphate, and intact PTH every 6 months in addition to baseline and 3 months. Majority of patients (95%) used D-Ca of 2.5 mEq/L or 3.0 mEq/L, and only the small proportion of patients used D-Ca of 2.75 mEq/L or 2.9 mEq/L. Therefore, comparison was done between the two groups of D-Ca of 2.5 and 3.0 mEq/L.

In the original J-DAVID trial, the primary outcome was the composite of fatal and non-fatal cardiovascular events (N = 188), and the secondary outcome was all-cause mortality (N = 169). To increases the power of statistical analysis, this post hoc analysis additionally considered the combined outcome including both the primary and secondary outcomes (N = 277).

To examine the difference in the effect of the alfacalcidol use on the clinical events between the D-Ca on the study enrollment, we conducted multivariable Cox proportional hazard regression analyses considering the time-varying treatment. The regression models included a cross-product term between the treatment and the baseline D-Ca value. The estimators were adjusted for the baseline covariates (sex, age, vintage of dialysis, and history of CVD, diabetic kidney disease, body mass index, C-reactive protein, serum levels of albumin and phosphate, corrected calcium, intact PTH, HDL cholesterol, hemoglobin, systolic blood pressure, and use of intravenous iron) and the time-varying covariates (body mass index, C-reactive protein, serum levels of albumin, phosphate, corrected calcium, intact PTH, hemoglobin, HDL-cholesterol, use of intravenous iron, systolic blood pressure).

The distributions of the covariates on the baseline were shown in the original paper [4]. To convert serum level expressed mg/dL to mmol/L, multiply by 0.25 for serum calcium and 0.323 for serum phosphate.

The J-DAVID trial was conducted in accordance with the principles of the Declaration of Helsinki and the Ethical Guidelines for Clinical Studies by the Ministry of Health, Labor and

Welfare, Japan (the original 2003 version, which was modified in 2004 and 2006). The protocol and revisions of this trial were approved by the ethics committee at the Osaka City University Graduate School of Medicine in Japan (approval numbers 1227, 1297, 1385, and 1525) and by the relevant ethics committees or institutional review boards at the study sites. All participants gave written informed consent before the study. The protocol of post hoc analyses using the original J-DAVID trail was reviewed and approved by the ethics committee of the Osaka City University Graduate School of Medicine (Number 4420 on September 26, 2019).

## Results

Data of predefined laboratory abnormalities is available in eTable 4 in the original paper [9]. In Table 1 and Fig 1, the trends of serum levels of corrected Ca, intact PTH, phosphate, and in both D-Ca 2.5 and 3.0 mEq/L were shown. Although all three variables were within the target ranges in most patients, the intervention group showed a temporal increase in corrected Ca and a decrease in intact PTH after starting alfacalcidol, and these changes returned to the initial levels thereafter. These changes were similarly observed in both D-Ca 2.5 and 3.0 mEq/L users.

In the stratified analyses by D-Ca, treatment with alfacalcidol was not significantly associated with the composite cardiovascular events, all-cause mortality, or the combined outcome

**Table 1. Prevalence of laboratory abnormalities by allocation and dialysate calcium concentration at each study month.**

| Abnormalities | Allocation | D-Ca (mE/L) | Baseline | 3M | 6M | 12M | 18M | 24M | 30M | 36M | 42M | 48M |
|---|---|---|---|---|---|---|---|---|---|---|---|---|
| Corrected Calcium > 10.0 mg/dL | Intervention | 2.5 | 1.5% | 23.0% | 19.2% | 12.8% | 10.1% | 12.1% | 9.0% | 8.5% | 11.2% | 11.8% |
| | | 3.0 | 4.5% | 31.5% | 23.4% | 17.1% | 13.7% | 10.4% | 13.6% | 10.2% | 9.9% | 8.2% |
| | Control | 2.5 | 0.8% | 1.7% | 2.6% | 7.2% | 5.5% | 4.8% | 7.0% | 3.1% | 2.2% | 6.0% |
| | | 3.0 | 5.7% | 8.3% | 6.9% | 8.7% | 10.8% | 8.5% | 11.4% | 10.5% | 11.3% | 6.9% |
| Corrected Calcium > 11.0 mg/dL | Intervention | 2.5 | 0.0% | 1.6% | 3.2% | 0.9% | 0.9% | 0.0% | 1.0% | 3.2% | 2.2% | 1.2% |
| | | 3.0 | 0.0% | 7.2% | 3.5% | 2.6% | 2.4% | 1.1% | 0.4% | 1.2% | 0.4% | 0.4% |
| | Control | 2.5 | 0.0% | 0.8% | 0.0% | 0.9% | 1.8% | 1.0% | 1.0% | 1.0% | 0.0% | 0.0% |
| | | 3.0 | 0.0% | 0.9% | 1.3% | 0.3% | 0.3% | 1.4% | 0.4% | 0.8% | 0.8% | 0.9% |
| Phosphate > 6.0 mg/dL | Intervention | 2.5 | 0.0% | 23.8% | 20.8% | 18.8% | 19.3% | 15.0% | 22.0% | 19.1% | 12.4% | 12.9% |
| | | 3.0 | 0.0% | 17.1% | 19.9% | 23.7% | 20.1% | 18.6% | 21.5% | 17.2% | 19.8% | 22.1% |
| | Control | 2.5 | 0.0% | 17.8% | 17.1% | 14.4% | 14.7% | 14.4% | 19.0% | 21.9% | 15.2% | 21.4% |
| | | 3.0 | 0.0% | 13.2% | 11.0% | 16.2% | 13.6% | 18.8% | 16.2% | 20.7% | 19.4% | 20.7% |
| Phosphate > 7.0 mg/dL | Intervention | 2.5 | 0.0% | 10.3% | 6.4% | 4.3% | 7.3% | 2.8% | 6.0% | 6.4% | 4.5% | 4.7% |
| | | 3.0 | 0.0% | 5.6% | 6.0% | 6.2% | 5.5% | 5.0% | 5.7% | 5.1% | 3.7% | 7.4% |
| | Control | 2.5 | 0.0% | 7.6% | 6.0% | 6.3% | 5.5% | 1.9% | 9.0% | 6.2% | 5.4% | 7.1% |
| | | 3.0 | 0.0% | 1.5% | 1.3% | 6.2% | 4.4% | 6.4% | 3.3% | 5.5% | 4.0% | 5.2% |
| Intact PTH > 240 pg/mL | Intervention | 2.5 | 0.0% | 1.6% | 0.8% | 2.7% | 0.9% | 9.4% | 6.1% | 11.0% | 12.6% | 14.6% |
| | | 3.0 | 0.0% | 0.4% | 2.2% | 1.9% | 5.3% | 2.0% | 6.2% | 9.1% | 10.9% | 15.3% |
| | Control | 2.5 | 0.0% | 4.4% | 6.3% | 13.6% | 17.3% | 22.5% | 15.6% | 13.0% | 14.8% | 16.0% |
| | | 3.0 | 0.0% | 4.7% | 4.1% | 6.4% | 6.6% | 8.4% | 11.4% | 13.1% | 11.7% | 12.9% |
| Intact PTH > 500 pg/mL | Intervention | 2.5 | 0.0% | 0.0% | 0.0% | 0.0% | 0.0% | 0.9% | 1.0% | 1.1% | 0.0% | 1.2% |
| | | 3.0 | 0.0% | 0.0% | 0.0% | 0.0% | 0.4% | 0.0% | 0.4% | 0.0% | 0.5% | 0.9% |
| | Control | 2.5 | 0.0% | 0.0% | 0.0% | 0.0% | 0.0% | 1.0% | 0.0% | 0.0% | 1.1% | 0.0% |
| | | 3.0 | 0.0% | 0.0% | 0.0% | 0.0% | 0.0% | 0.8% | 0.0% | 0.4% | 0.0% | 0.0% |

The prevalence of each laboratory abnormality was calculated by the number of patients with the abnormality divided by the number of patients with the measurement at each study month. Abbreviations: D-Ca, dialysate calcium; M, month; PTH, parathyroid hormone.

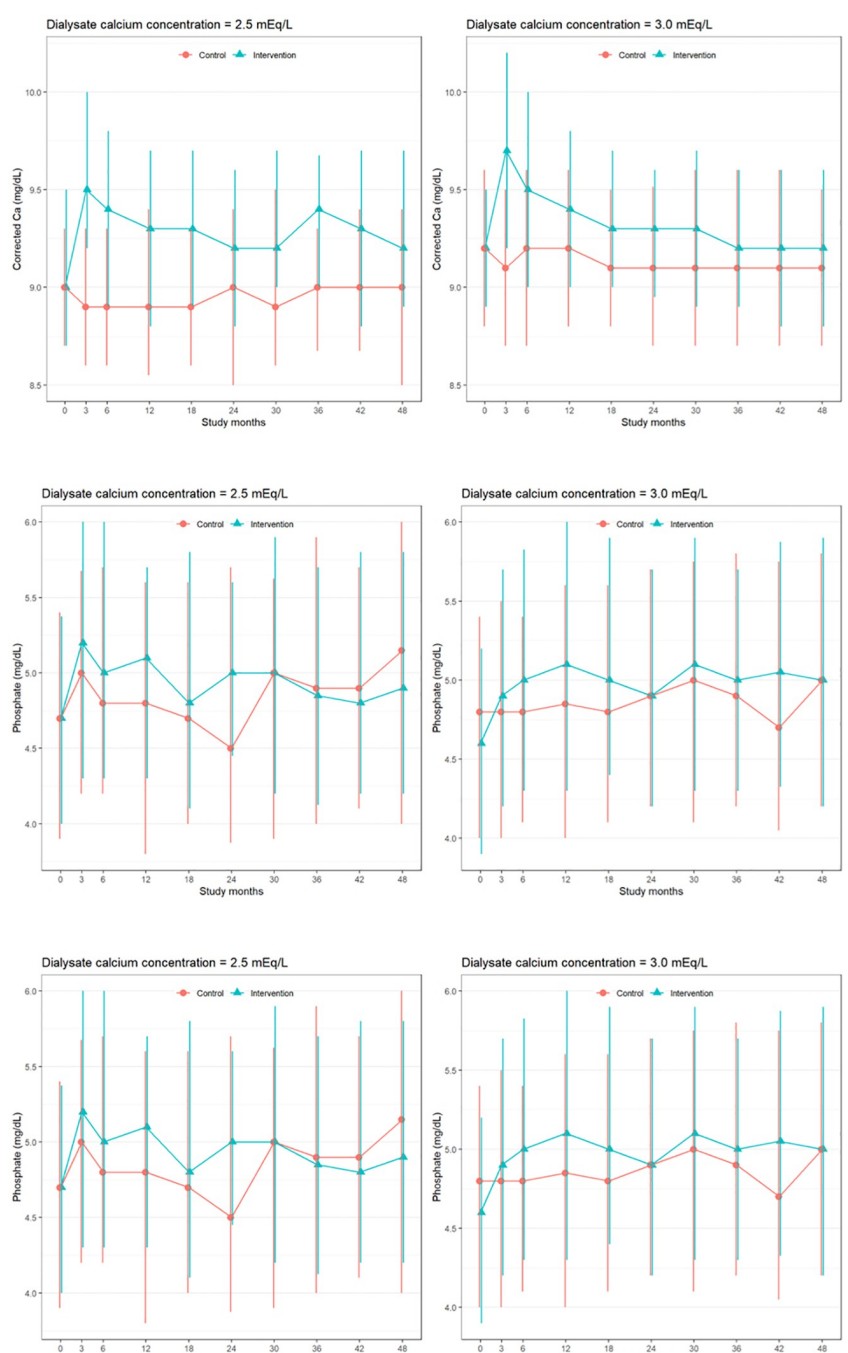

**Fig 1. Serial changes in the median (25th, 75th percentile) of serum levels of corrected Ca, phosphate, and intact PTH in D-Ca 2.5 and 3.0 mEq/L.**

in either D-Ca concentration (Table 2). The associations of treatment with alfacalcidol with these outcomes were not significantly modified by D-Ca (Fig 2).

Also, in the stratified analyses by treatment with alfacalcidol, the D-Ca was not significantly associated with the composite cardiovascular events, all-cause mortality, or the combined

**Table 2. Estimated association of alfacalcidol use with the two outcomes stratified by dialysate calcium concentration.**

| Exposure | Outcome | Strata | Hazard Ratio (95% CI) | P-value |
|---|---|---|---|---|
| Alfacalcidol (use vs. nonuse) | Composite CVD events | D-Ca 2.5 mEq/L | 1.119 (0.626–1.997) | 0.705 |
| | | D-Ca 3.0 mEq/L | 1.066 (0.703–1.619) | 0.762 |
| | ACM | D-Ca 2.5 mEq/L | 1.047 (0.523–2.099) | 0.896 |
| | | D-Ca 3.0 mEq/L | 0.710 (0.440–1.147) | 0.161 |
| | Composite CVD events or ACM | D-Ca 2.5 mEq/L | 1.087 (0.651–1.815) | 0.749 |
| | | D-Ca 3.0 mEq/L | 0.910 (0.638–1.298) | 0.602 |

Abbreviations: CVD, cardiovascular disease; ACM, all-cause mortality; D-Ca, dialysate calcium concentration; CI, confidence interval.

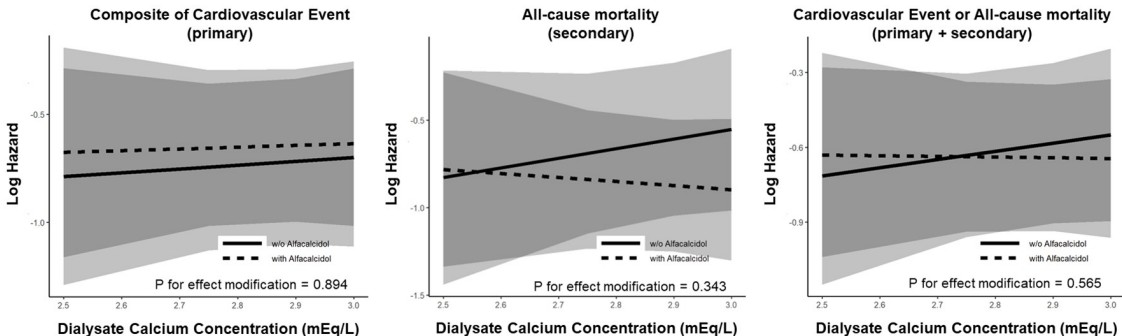

**Fig 2. Estimated hazards of composite cardiovascular events (primary outcome), all-cause mortality (secondary outcome), and the combined outcome (primary plus secondary) by the use of alfacalcidol and the baseline dialysate calcium concentration.** Estimated hazard was plotted against the dialysate calcium concentration stratified by the use of alfacalcidol.

outcome regardless of treatment with alfacalcidol (Table 3). The associations of D-Ca with these clinical outcomes were not significantly modified by the use of alfacalcidol (Fig 2).

## Discussion

The present study showed that the D-Ca in the range of 2.5 to 3.0 mEq/L did not have statistically significant impact on the primary and secondary outcomes of the J-DAVID participants in either the users or non-users of alfacalcidol. Other than D-Ca, many factors such as medication, diet, bone status, and ultrafiltration volume are involved in the calcium balance of HD patients. Optimal D-Ca has been discussed before [11, 12] and after [13, 14] the introduction of calcimimetics (calcium-sensing receptor agonists). In the recent KDIGO CPG, D-Ca with

**Table 3. Estimated association of dialysate calcium concentration with the two outcomes stratified by use of alfacalcidol.**

| Exposure | Outcome | Strata | Hazard Ratio (95% CI) | P-value |
|---|---|---|---|---|
| D-Ca (3.0 vs 2.5 mEq/L) | Composite CVD events | Alfacalcidol use (−) | 1.093 (0.665–1.795) | 0.727 |
| | | Alfacalcidol use (+) | 1.042 (0.636–1.705) | 0.871 |
| | ACM | Alfacalcidol use (−) | 1.315 (0.735–2.353) | 0.356 |
| | | Alfacalcidol use (+) | 0.891 (0.507–1.566) | 0.689 |
| | Composite CVD events or ACM | Alfacalcidol use (−) | 1.179 (0.763–1.821) | 0.458 |
| | | Alfacalcidol use (+) | 0.986 (0.649–1.499) | 0.949 |

Abbreviations: CVD, cardiovascular disease; ACM, all-cause mortality; D-Ca, dialysate calcium concentration; CI, confidence interval.

the range of 2.5 to 3.0 mEq/L was suggested [1, 15]. Within this range of D-Ca, the effect of alfacalcidol on clinical outcomes was not significantly modified.

One may speculate that treatment with VDRA is harmful in patients without secondary hyperparathyroidism who are dialyzed against higher concentrations of D-Ca because calcium load is higher. However, we noticed a numerically higher hazard for the composite of cardio-vascular events associated with a higher D-Ca in patients who were not treated with alfacalci-dol, not in patients who were treated with alfacalcidol, although the result was not statistically significant. The same was true in the association of D-Ca with the hazard for all-cause mortal-ity. Although the exact mechanisms for these observations are unknown, the calcium load from dialysate may not have a detectable impact on clinical outcomes in the range of 2.5 to 3.0 mEq/L of D-Ca. According to the JSDT data in 2009, D-Ca was 2.5 to 3.0 mEq/L in more than 96% of the total number of reported dialysis patients (N = 209,322) in Japan. Otherwise, the result might have been affected the fact that intravenous VDRAs other than oral alfacalcidol were allowed to be used if needed to follow the JSDT guideline, and that more patients in the control group received intravenous VDRAs than the intervention group of the original J-DAVID trial.

This *post hoc* analysis was not able to show statistically significant association of D-Ca with composite cardiovascular events or all-cause mortality. The lack of significant associations may be attributable to the relatively low statistical power of the original J-DAVID trial. To address this issue, we took another outcome including both the primary composite cardio-vascular outcome and the secondary all-cause mortality in this *pos hoc* analysis. However, the association of D-Ca and the combined outcome was again not significant. Similar to our results, another observational study derived from the Japanese Dialysis Outcomes and Practice Patterns Study reported that the difference in D-Ca was not related to the survival based on the results of 9,201 patients with a median follow-up of 2.03 years [16]. Thus, the low statistical power does not fully explain the neutral results on the D-Ca in this range and clinical out-comes in this study.

Kim HW et al reported that high D-Ca (3.5 mEq/L) was associated with a higher mortality rate and hospitalization with CVD or infection among incident HD patients [17]. *"There are potential safety concerns associated with the default use of dialysate calcium concentrations (2.50 mEq/L)"* [18]. High D-Ca (3.5 mEq/L) has a higher risk of coronary artery calcification than that lower D-Ca (2.5 mEq/L) [19]. Tagawa et al reported that High (3.0 mEq/L and over) was associated with incident MI among DM dialysis patients with low bone turnover [20].

Control of serum levels of Ca, phosphate, and PTH is mandatory for the prevention and management of CKD-MBD. Hyperphosphatemia is independently associated with an elevated risk of sudden death in patients on hemodialysis [21]. Yamada S et al reported that high serum calcium (10.0–16.5 mg/dL) is a risk factor of infection-related and all-cause death in hemodial-ysis patients [22]. Concerning the treatment of hyperphosphatemia either non-Ca-based or Ca-based phosphate binder, Ogata H et al compare the effects between non-calcium-based phosphate binders and calcium-based binders for reducing cardiovascular events. There was no difference in all-cause mortality among Japanese HD patients. However, the results may be confounded by the relatively low incidence of cardiovascular events among Japanese patients [23].

Sakoh T et al conducted a short-term (6 months) intervention study to see the effect of changing D-Ca, from 3.0 to 2.75 mEq/L (N = 12) and 2.5 to 2.75 mEq/L (N = 12) [2]. Although the intradialytic Ca loading was different, there were no significant differences in serum Ca, phosphate, PTH, and fibroblast growth factor 23. Yamada S et al observed the effects of lower-ing D-Ca from 3.0 to 2.75 mEq/L [24]. One year after the conversion, the mean serum Ca level decreased, while serum phosphate, alkaline phosphatase, and PTH increased. These authors

concluded that D-Ca should be individualized based on clinical factors. However, a single patient dialysis fluid delivery system (SPDDS) is very uncommon in Japan as the central dialysis fluid delivery system (CDDS) is safe and evolved for more than 50 years [3].

Calcimimetic agents (calcium-sensing receptor agonists) have been available since January 2008 in Japan. During the recruiting period, July 1, 2008, to January 26, 2011, cinacalcet (the first calcimimetic agent in the Japanese market) was used at the start of the J-DAVID trial in 5.9% of the total participants [9]. Most of the dialysis units followed the JSDT guideline published in 2008 [25] and updated in 2013 accordingly [26]. Both cinacalcet and evocalcet are effective for controlling secondary hyperparathyroidism [27] irrespective of D-Ca [28, 29]. By comparing the effects on serum calcification propensity (T50), a surrogate marker of calcification, Shoji T et al reported that a calcimimetic agent etelcalcetide was more effective than a vitamin D receptor activator maxacalcitol [30].

Regular monitoring of serum levels of calcium, phosphate, and PTH is mandatory for the control of CKD-MBD and improving survival. Large differences in these parameters among countries and races, yet the potential confounders of these observations remained to be studied. [31]. However, precise control of serum phosphate and calcium may achieve adequate levels of PTH [32–35]. The target of PTH is low as 60 to 180 pg/ml and the survival is excellent in Japan [36].

The strengths of the present study are the post hoc analysis of the J-DAVID trial; therefore, the observation period is relatively long, with a median of 4 years, and the follow-up rate was quite high. Serial laboratory data on mineral metabolism such as corrected Ca, phosphate, and intact PTH were available. The rates of exclusion and loss to follow-up were small as 1.2% and 2.0%, respectively. However, there are several limitations in the present study. Firstly, the study subjects were those with no evidence of severe secondary hyperparathyroidism whose baseline intact PTH levels were 180 pg/mL or lower. Therefore, we could not examine the possible effects of alfacalcidol and different D-Ca levels and the interactions on the clinical outcomes. Second, the vintage of HD was relatively short as the median of 5.6 years. Third, dialysate composition other than calcium was not considered. Finally, the statistical power to detect the differences if any was small as stated in the original paper [4]. This is partly due to the number of events being small.

In conclusion, we observed no significant effects of D-Ca on the CKD MBD parameters and the clinical outcomes. Results support the notion that the current strategy of using D-Ca between 2.5 to 3.0 mEq/L is safe. Further research is necessary for those with secondary hyperparathyroidism.

## Supporting information

**S1 File.**
(CSV)

## Acknowledgments

We thank all the collaborators for the J-DAVID trial. A list of them is available in the original paper [4].

## Author Contributions

**Conceptualization:** Kunitoshi Iseki, Tetsuo Shoji, Masaaki Inaba.

**Data curation:** Daijiro Kabata.

**Funding acquisition:** Tetsuo Shoji.

**Investigation:** Kunitoshi Iseki, Daijiro Kabata.

**Methodology:** Kunitoshi Iseki.

**Supervision:** Tetsuo Shoji.

**Visualization:** Daijiro Kabata.

**Writing – original draft:** Kunitoshi Iseki.

**Writing – review & editing:** Kunitoshi Iseki, Daijiro Kabata, Tetsuo Shoji, Masaaki Inaba, Masanori Emoto, Katsuhito Mori, Tomoaki Morioka, Shinya Nakatani, Ayumi Shintani.

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
