## [Decision Letter · Decision Letter 0]

31 Mar 2022

PONE-D-22-04275Dialysate Calcium, Alfacalcidol, and clinical Outcomes: A Post-Hoc Analysis of the J-DAVID TrialPLOS ONE

Dear Dr. Iseki,

Thank you for submitting your manuscript to PLOS ONE. After careful consideration, we feel that it has merit but does not fully meet PLOS ONE’s publication criteria as it currently stands. Therefore, we invite you to submit a revised version of the manuscript that addresses the points raised during the review process.

We look forward to receiving your revised manuscript.

Kind regards,

Larry Allan Weinrauch, MD

Academic Editor

PLOS ONE

Journal Requirements:

"Kunitoshi Iseki received personal fees from Kyowa Kirin, Bayer Yakuhin, Torii Pharmaceutical, Mochida Pharmaceutical, and Kissei Pharmaceutical; and received consultation fee from the MediGate Digital Health K.K., Daijiro Kabata received personal fees from Chugai Pharmaceutical; research grant and consultation fee for statistical analysis from Kyowa Kirin; and personal fees and a research grant from Bayer Yakuhin outside the submitted work. Ayumi Shintani received personal fees from Chugai Pharmaceutical, Bayer Yakuhin, and Ono Pharmaceutical; and personal fees and research grants from Kyowa Kirin, Takeda Pharmaceutical, and Daiichi Sankyo outside the submitted work. Shinya Nakatani received personal fees from Chugai Pharmaceutical, Kyowa Kirin, Bayer Yakuhin, Ono Pharmaceutical, Kissei Pharmaceutical, and Torii Pharmaceutical. Tomoaki Morioka received personal fees from Ono Pharmaceutical. Katsuhito Mori received personal fees from Ono Pharmaceutical; he served as the principal investigator of a clinical trial for Kyowa Kirin; and personal fees and a research grant from Mitsubishi Tanabe outside the submitted work. Masaaki Inaba received personal fees from Chugai Pharmaceutical, Kyowa Kirin, Bayer Yakuhin, Ono Pharmaceutical, Kissei Pharmaceutical, and Torii Pharmaceutical outside the submitted work. Masanori Emoto received personal fees from Kissei Pharmaceutical, AstraZeneca, and Torii Pharmaceutical; and personal fees and research grants from Chugai Pharmaceutical, Kyowa Kirin, Ono Pharmaceutical, Nippon Boehringer Ingelheim, and Mitsubishi Tanabe outside the submitted work. Tetsuo Shoji received personal fees from Chugai Pharmaceutical, Kyowa Kirin, and Kissei Pharmaceutical; and personal fees and research grants from Bayer Yakuhin, and Ono Pharmaceutical outside the submitted work."

5. Please include your tables as part of your main manuscript and remove the individual files. Please note that supplementary tables (should remain/ be uploaded) as separate "supporting information" files"

Additional Editor Comments:

Thank you for your submission.

The reviewers pointed out a few minor changes that would improve this manuscript and I hope that you are able to respond to their constructive tenor.

Reviewers' comments:

Reviewer's Responses to Questions

**Comments to the Author**

1. Is the manuscript technically sound, and do the data support the conclusions?

Reviewer #1: Yes

Reviewer #2: Yes

2. Has the statistical analysis been performed appropriately and rigorously? 

Reviewer #1: Yes

Reviewer #2: Yes

3. Have the authors made all data underlying the findings in their manuscript fully available?

Reviewer #1: Yes

Reviewer #2: Yes

4. Is the manuscript presented in an intelligible fashion and written in standard English?

Reviewer #1: Yes

Reviewer #2: Yes

5. Review Comments to the Author

Reviewer #1: Another nice publication out of J-DAVID trial.

1. As with the previous J-DAVID publication in JAMA (JAMA. 2018;320(22):2325–2334), I have difficulty understanding the definition of "HD patients without secondary hyperparathyroidism" that they chose "PTH of less than or equal to 180 pg/ml."

Although we typically do not use VDRA in those patients, the PTH of 180 pg/ml is close to 3times upper limit of normal and in most labs PTH of > 80 pg/ml is considered secondary hyperparathyroidism. A better term might be mild secondary hyperparathyroidism that guidelines does not require VDRA treatment.

In the previous JAM article they mention: "...Compared with the control group, the intervention group had lower proportions of calcium carbonate users and cinacalcet users, and higher proportions of sevelamer users and lanthanum carbonate users. The proportion of intravenous VDRA users increased over time in both groups, although the proportion was higher in the control group." That raises many question about the study for me:

2. Why did they used IV VDRA or cinacalcet (I assume much rise in PTH), and what happened to comparisons in those who did get IV VDRA and those who did not?

3. The main reason most guideline, including KDIGO avoid lowering PTH more, and use of VDRA is not required in PTH of <180 pg/ml, is to avoid the low bone turnover and dynamic bone which is a disaster to treat. Higher calcium and and even lower PTH were seen initially in both high D-Ca groups and VDRA users in J-DAVID trial. Did they look into adverse effects of potential low bone turnover in these groups?

Reviewer #2: The study by Iseki et al is a very nice post-hoc analysis of the J-DAVID trial is very nicely written. The authors compared 2.5 and 3 mEq/L of dialysate calcium and hwo they responded to treatment verse control with VDRA. I have a couple concerns about the study that I would like the authors to address:

1) The authors did not discuss the power of seeing a difference with treatment. The original study reported that you needed a sample size of 1600 to see a difference between groups with an expected primary outcome of 32% in the control group. However the authors only had 333/331 individuals in the 3.0 mEq/L Ca. However, for the primary outcome of mortality the 3.0 group had a hazard ratio of 0.710 with a p value of 0.16. The authors should discuss the fact that intervention may have an affect on mortality in those dialyzed against a higher Ca, but they may have been underpowered.

2) In table 1a it is confusing. Instead of putting in Number of reported values under abnormalities. Maybe change it to all reported calcium, PTH and phosphate?

Minor concerns:

1) Line 32 please change oral alfacalcidol to vitamin D receptor agonist (oral alfacidol)

2) Line 56 please change vitamin receptor agonist to vitamin D receptor agonist

3) Last line page 3, line 77-78. Needs to be reworded.

4) Result section should all be changed to present tense rather than past tense.

5) Line 112 the he should be changed to the

6. PLOS authors have the option to publish the peer review history of their article (what does this mean?). If published, this will include your full peer review and any attached files.

Reviewer #1: **Yes: **Bijan Roshan

Reviewer #2: No

---

## [Author Response · Author response to Decision Letter 0]

18 May 2022

Response to comments by the Editor and the two Reviewers

1. Our response to comments by the editor

We appreciate the time and effort of the editor in evaluating our manuscript.

Editor comment #1. 

Our reply

We did it as required. We have combined the files for each part of the figure, and now, the file names for figures are Fig1.pptx, Fig2.pptx. Because it is difficult for us to convert file format, can you kindly convert these pptx files into the format appropriate for PLOS One, please?

Editor comment #2. 

Please provide additional details regarding participant consent. In the ethics statement in the Methods and online submission information, please ensure that you have specified (1) whether consent was informed and (2) what type you obtained (for instance, written or verbal, and if verbal, how it was documented and witnessed). If your study included minors, state whether you obtained consent from parents or guardians. If the need for consent was waived by the ethics committee, please include this information.

Our reply

All the participants of the J-DAVID trial gave written informed consent. This has been indicated in the revised manuscript. In this revision, we would like to rewrite this part in order to mention the approvement of the original J-DAVID trial by ethics committee. Please see Change #5 in the list of changes below. 

Editor comment #3. 

We note that the grant information you provided in the ‘Funding Information’ and ‘Financial Disclosure’ sections do not match. 

Our reply

The information in ‘Financial Disclosure’ section is correct, and it has not been changed during this revision. This study and the original J-DAVID trial were supported by the grant to TS from The Kidney Foundation, Japan as indicated in the original version of this manuscript. The grant number was not indicated, because this grant had no specific grant number. Therefore, in response to the above comment, we have added that this grant had no specific number (Change #11). Some coauthors disclosed research grant, but all these grants were not related to the J-DAVID trial or the post hoc analysis. 

 

Editor comment #4. 

Thank you for stating the following in the Competing Interests section: 

"Kunitoshi Iseki received personal fees from Kyowa Kirin, Bayer Yakuhin, Torii Pharmaceutical, Mochida Pharmaceutical, and Kissei Pharmaceutical; and received consultation fee from the MediGate Digital Health K.K., Daijiro Kabata received personal fees from Chugai Pharmaceutical; research grant and consultation fee for statistical analysis from Kyowa Kirin; and personal fees and a research grant from Bayer Yakuhin outside the submitted work. Ayumi Shintani received personal fees from Chugai Pharmaceutical, Bayer Yakuhin, and Ono Pharmaceutical; and personal fees and research grants from Kyowa Kirin, Takeda Pharmaceutical, and Daiichi Sankyo outside the submitted work. Shinya Nakatani received personal fees from Chugai Pharmaceutical, Kyowa Kirin, Bayer Yakuhin, Ono Pharmaceutical, Kissei Pharmaceutical, and Torii Pharmaceutical. Tomoaki Morioka received personal fees from Ono Pharmaceutical. Katsuhito Mori received personal fees from Ono Pharmaceutical; he served as the principal investigator of a clinical trial for Kyowa Kirin; and personal fees and a research grant from Mitsubishi Tanabe outside the submitted work. Masaaki Inaba received personal fees from Chugai Pharmaceutical, Kyowa Kirin, Bayer Yakuhin, Ono Pharmaceutical, Kissei Pharmaceutical, and Torii Pharmaceutical outside the submitted work. Masanori Emoto received personal fees from Kissei Pharmaceutical, AstraZeneca, and Torii Pharmaceutical; and personal fees and research grants from Chugai Pharmaceutical, Kyowa Kirin, Ono Pharmaceutical, Nippon Boehringer Ingelheim, and Mitsubishi Tanabe outside the submitted work. Tetsuo Shoji received personal fees from Chugai Pharmaceutical, Kyowa Kirin, and Kissei Pharmaceutical; and personal fees and research grants from Bayer Yakuhin, and Ono Pharmaceutical outside the submitted work."

Our reply

1We have added the statement: "This does not alter our adherence to PLOS ONE policies on sharing data and materials” in the Competing Interests section （Change #12）. Data sharing statement is also added （Change #13）. The competing interests information written in the manuscript is correct and no update is needed. As suggested, we have included the Competing interests information in the cover letter. Can you handle it appropriately in the editorial office, please?

Editor comment #5. 

Please include your tables as part of your main manuscript and remove the individual files. Please note that supplementary tables (should remain/ be uploaded) as separate "supporting information" files"

Our reply

We have included our tables as part of our main manuscript （Changes #14, #15）and removed the individual files which were originally submitted in PPT files. Also, supplementary tables have been handled as suggested. 

Editor comment #6. 

Our reply

We have reviewed the reference list and confirmed that it is correct. No change has been made to it during revision.

 

2. Our response to comments by Reviewer #1

We appreciate the time and effort of the reviewer in evaluating our manuscript.

Reviewer #1, Comment #1 

Another nice publication out of J-DAVID trial.

As with the previous J-DAVID publication in JAMA (JAMA. 2018;320(22):2325–2334), I have difficulty understanding the definition of "HD patients without secondary hyperparathyroidism" that they chose "PTH of less than or equal to 180 pg/ml."

Although we typically do not use VDRA in those patients, the PTH of 180 pg/ml is close to 3times upper limit of normal and in most labs PTH of > 80 pg/ml is considered secondary hyperparathyroidism. A better term might be mild secondary hyperparathyroidism that guidelines does not require VDRA treatment.

In the previous JAM article they mention: "...Compared with the control group, the intervention group had lower proportions of calcium carbonate users and cinacalcet users, and higher proportions of sevelamer users and lanthanum carbonate users. The proportion of intravenous VDRA users increased over time in both groups, although the proportion was higher in the control group." That raises many question about the study for me:

Our reply . 

This comment includes two different points, namely, 1-1. The term which defined the participants who had intact PTH of 180 pg/mL or lower, and 1-2. The potential problems which were introduced by the use of IV-VDRA in the two groups in the original J-DAVID trial. 

1-1. We agree that the term “secondary hyperparathyroidism” used in the original J-DAVID trial may be somewhat confusing. According to the 2006 JSDT guideline, the recommended range of intact PTH was 60-180, which was higher than the reference range of intact PTH which is set for “normal” individuals not on dialysis. However, although the target range of intact PTH was set by the JSDT guideline and other clinical practice guidelines, there is no clear DEFINITION of secondary hyperparathyroidism (SHPT) for patients on dialysis. Therefore, we used the term SHPT in the original JAMA paper to mean “elevated intact PTH which needs treatment” namely PTH levels over the target range. In the clinical practice of hemodialysis patients, PTH level in this range was considered appropriate and NOT regarded as “hyperparathyroidism”. The use of this term was not criticized by any reviewers or editors of JAMA. Therefore, although we agree that it may be somewhat confusing, we do not believe that the term is not wrong. 

In response to this comment, we have avoided the expression of “patients without secondary hyperparathyroidism” in the Abstract. Please see Change #1 in the list of changes below.

1-2. We appreciate this question. We would like to point out the difficulty in clinical trials in CKD-MBD. There are many players in CKD-MBD such as calcium, phosphate, PTH, vitamin D, FGF23, and others. All of these are closely linked in a complex manner. Therefore, if one of these factors is intentionally changed in a clinical trial, other components will change accordingly. Therefore, it was ethically needed to permit the use of some drugs other than the test drug alfacalcidol. In cases in whom SHPT developed even treated with alfacalcidol, it may be needed to change alfacalcidol to more potent intravenous VDRA to follow the clinical practice guideline. This was also the case with the patients in the control group. If no VDRA was given, secondary hyperparathyroidism would have been more advanced as its natural course. Thus, some rescues other than the test drug were ethically needed in the control group. This difficult situation of CKD-MBD trials is quite different from that of trials with a statin. 

When the J-DAVID trial was open, cinacalcet was not available in Japan. It was used during the recruitment period used in Japan. Therefore, investigators in each site used IV VDRA to control SHPT if needed. 

Therefore, the J-DAVID trial could not evaluate the pure pharmacological effect of alfacalcidol, but the results of the J-DAVID indicate that the “VDRA first” strategy did not result in better clinical outcomes in hemodialysis patients with intact PTH of 180 pg/mL or lower as compared to the “VDRA last” strategy. 

Since we supposed the use of IV-VDRA in the J-DAVID trial, we pre-planned analyses which considered the use of IV-VDRA, in addition to the primary ITT analysis using the full analysis set (FAS). Namely, analyses using per-protocol set (PPS) and modified per-protocol set (m-PPS). In PPS, patients who switched from oral alfacalcidol to IV-VDRA in the intervention group were censored, and patients who used IV-VDRA in the control group were censored, at the time of IV-VDRA use. In m-PPS, patients who used IV-VDRA in the control group were similarly censored at the time of IV-VDRA use, whereas patients who switched from oral alfacalcidol to IV-VDRA in the intervention group were not censored because they were kept treated with any VDRA. Very interestingly, these analyses with PPS and m-PPS showed higher hazard ratios than the primary analysis, and the P-values were lower. The analysis with m-PPS gave HR of 1.36 (0.99-1.87), P=0.06, when adjusted for age, sex, diabetic kidney disease, dialysis vintage, prior CVD, and facilities (Please see Table 3 of the JAMA paper). 

Thus, the use of IV-VDRA in the control group introduced contamination of treatment in the trial and reduced the statistical power of this trial. When the effect of contamination was carefully removed, the use of alfacalcidol and other IV-VDRA appeared to increase the risk of CVD in this population, although it was not statistically significant. 

In response to this comment, the possible influence of IV-VDRA used in the control group has been discussed in the revised manuscript (Change #10). 

Reviewer #1, Comment #2. 

Why did they used IV VDRA or cinacalcet (I assume much rise in PTH), and what happened to comparisons in those who did get IV VDRA and those who did not?

Our reply

We appreciate this comment. According to the protocol of the original J-DAVID trial, IV-VDRAs other than the test drug (oral alfacalcidol) were allowed to be used if needed to follow the JSDT guideline. Therefore, although we recorded the exact reason for the use of IV-VDRA, we also assume that these patients experienced an increase in intact PTH which needed intensified medical treatment. Although we did not compare the clinical outcomes between those who took IV-VDRA and those who did not, the possible effects of the use of IV-VDRA in the control group and in the intervention group were addressed in additional analyses using PPS and m-PPS as mentioned above. In response to this comment, we have mentioned the possible reason for the use of IV-VDRA in the revised manuscript (Change #10).

Reviewer #1, Comment #3. 

The main reason most guidelines, including KDIGO avoid lowering PTH more and use of VDRA is not required in PTH of <180 pg/ml, is to avoid the low bone turnover and dynamic bone which is a disaster to treat. Higher calcium and even lower PTH were seen initially in both high D-Ca groups and VDRA users in J-DAVID trial. Did they look into the adverse effects of potential low bone turnover in these groups?

Our reply

We appreciate this comment. We did not examine whether bone turnover affected the clinical outcome in the original J-DAVID trial. However, bone turnover may modify the effect of alfacalcidol on the clinical outcomes as pointed out by the reviewer. Therefore, we have performed another post hoc analysis using ALP activity as an index of bone turnover, and we found that the effect modification was not statistically significant. The results are currently in the review process in another journal. 

In response to this comment, we did not make changes in this manuscript.

We appreciate again the careful evaluation and constructive comments by the reviewer. 

 

Our response to comments by Reviewer #2:

We appreciate the time and effort of the reviewer in evaluating our manuscript.

Reviewer #2, Comment #1

The study by Iseki et al is a very nice posthoc analysis of the J-DAVID trial is very nicely written. The authors compared 2.5 and 3 mEq/L of dialysate calcium and hwo they responded to treatment verse control with VDRA. I have a couple concerns about the study that I would like the authors to address:

1) The authors did not discuss the power of seeing a difference with treatment. The original study reported that you needed a sample size of 1600 to see a difference between groups with an expected primary outcome of 32% in the control group. However the authors only had 333/331 individuals in the 3.0 mEq/L Ca. However, for the primary outcome of mortality the 3.0 group had a hazard ratio of 0.710 with a p value of 0.16. The authors should discuss the fact that intervention may have an affect on mortality in those dialyzed against a higher Ca, but they may have been underpowered.

We appreciate this comment. We agree with the reviewer. The lack of sufficient statistical power was already mentioned as one of study limitations in the original manuscript. We have discussed this issue in a new paragraph in the revised manuscript. Please see Change #10. in the list of changes below. 

Reviewer #2, Comment #2

Table 1a it is confusing. Instead of putting in Number of reported values under abnormalities. Maybe change it to all reported calcium, PTH and phosphate?

We agree that Table 1a was confusing. The same is true for Table 1b, Table 1c, and Table 1d. There were so many numbers in the table which may not be necessary for the readers. In response to the advice to remove “Number of reported values” from the column of “abnormalities”, we have revised these tables to be more simple and more concise (Change #14). Similarly, Table 2a and Table 2b have been edited into Table 2 and Table 3 (Change #15). We believe that these tables have become much more readable after revision. 

Reviewer #2, Minor concern #1

Line 32 please change oral alfacalcidol to vitamin D receptor agonist (oral alfacidol)

We appreciate this comment. Although we used “vitamin D receptor agonist”, “vitamin D receptor activator” is more common for the full spelling of VDRA. Therefore, we have changed it to a vitamin D receptor activator (Change #1). 

Reviewer #2, Minor concern #2

Line 56 please change vitamin receptor agonist to vitamin D receptor agonist

We have corrected this part (Change #2). 

Reviewer #2, Minor concern #3

Last line page 3, lines 77-78. Needs to be reworded.

We corrected it as suggested. (Change #3)

Reviewer #2, Minor concern #4

The result section should all be changed to present tense rather than past tense.

We do not agree with this advice. It is standard to describe the results in the past tense rather than present tense. 

Reviewer #2, Minor concern #5

Line 112 the he should be changed to the

We appreciate your very careful evaluation. Our careless mistake "he" has been changed to "the". But this part of the text has been further revised to be more readable (Change #7). 

We appreciate again the careful evaluation and constructive comments by the reviewer. 

 

List of changes

Change #1

Original, Page 2, Abstract, Line 31-32:

The trial was an open-label randomized clinical trial including 976 HD patients without secondary hyperparathyroidism which compared the users of oral alfacalcidol and non-users over a median of 4 years.

Revised, Page, Line :

The trial was an open-label randomized clinical trial including 976 HD patients with intact PTH of 180 pg/mL or lower which compared the users of vitamin D receptor activator (oral alfacalcidol) and non-users over a median of 4 years.

Change #2 

Original, Page 3, Line 56-57:

Many observational studies showed that the use of vitamin receptor agonist (VDRA) was associated with a lower risk of all-cause mortality (4, 5),

Revised, Page 3, Line 56-57:

Many observational studies showed that the use of vitamin D receptor activator (VDRA) was associated with a lower risk of all-cause mortality (4, 5),

Change #3 

Original, Page 3, Line 77-80:

The number and percentage of patients were too small as 5% with D-Ca 2.75 mEq/L and 2.9 mEq/L. Therefore, we obtained additional data such as corrected calcium, intact PTH, and serum phosphate according to the D-Ca of 2.5 and 3.0 mEq/L.

Revised, Page 3-4, Line 78-80:

Majority of patients (95%) used D-Ca of 2.5 mEq/L or 3.0 mEq/L, and only the small proportion of patients used D-Ca of 2.75 mEq/L or 2.9 mEq/L. Therefore, comparison was done between the two groups of D-Ca of 2.5 and 3.0 mEq/L.

Change #4

A new paragraph has been included in the Methods section, 2nd paragraph to explain the reason why the new combined outcome was considered. 

Revised, Page 4, Line 81-84:

In the original J-DAVID trial, the primary outcome was the composite of fatal and non-fatal cardiovascular events (N = 188), and the secondary outcome was all-cause mortality (N = 169). To increases the power of statistical analysis, this post hoc analysis additionally considered the combined outcome including both the primary and secondary outcomes (N = 277). 

Change #5 

Original, Page 4, Line 94-95:

The institutional review board of the Osaka City University approved the research project, Number 4420 on September 26, 2019.

Revised, Page 4-5, Line 98-106:

The J-DAVID trial was conducted in accordance with the principles of the Declaration of Helsinki and the Ethical Guidelines for Clinical Studies by the Ministry of Health, Labor and Welfare, Japan (the original 2003 version, which was modified in 2004 and 2006). The protocol and revisions of this trial were approved by the ethics committee at the Osaka City University Graduate School of Medicine in Japan (approval numbers 1227, 1297, 1385, and 1525) and by the relevant ethics committees or institutional review boards at the study sites. All participants gave written informed consent before the study. The protocol of post hoc analyses using the original J-DAVID trail was reviewed and approved by the ethics committee of the Osaka City University Graduate School of Medicine (Number 4420 on September 26, 2019).

Change #6 

Original, Page 4, Line 100-103:

In Table 1a, 1b, 1c, and 1d, the trends of serum levels of corrected Ca, intact PTH, phosphate, and in both D-Ca 2.5 and 3.0 mEq/L were shown. Figures 1a, 1b, and 1c showed the mean (SD) of corrected Ca, phosphate, and intact PTH in both D-Ca 2.5 and 3.0 mEq/L. Although all three variables were within the target ranges, …

Revised, Page 5, Line 111-113:

In Table 1 and Figure 1, the trends of serum levels of corrected Ca, intact PTH, phosphate, and in both D-Ca 2.5 and 3.0 mEq/L were shown. Although all three variables were within the target ranges in most patients, …

Change #7 

Original, Page 5, Line 107-114:

In the stratified analyses by D-Ca, the allocation group was not significantly associated with the composite of fatal and non-fatal cardiovascular events in either D-Ca concentration. Also, the D-Ca was not significantly associated with the composite of fatal and non-fatal cardiovascular events in either allocation group. (Table 2a). There were no significant differences between the groups by using interaction (Figure 2).

There were no significant differences between the groups by using interaction he risks hazards of all-cause mortality by D-Ca (Table 2b). There were no significant differences between the groups by using interaction (Figure 3).

Revised, Page 5, Line 116-123:

In the stratified analyses by D-Ca, treatment with alfacalcidol was not significantly associated with the composite cardiovascular events, all-cause mortality, or the combined outcome in either D-Ca concentration (Table 2). The associations of treatment with alfacalcidol with these outcomes were not significantly modified by D-Ca (Figure 2). 

Also, in the stratified analyses by treatment with alfacalcidol, the D-Ca was not significantly associated with the composite cardiovascular events, all-cause mortality, or the combined outcome regardless of treatment with alfacalcidol (Table 3). The associations of D-Ca with these clinical outcomes were not significantly modified by the use of alfacalcidol (Figure 2). 

Change #8 

Original, Page 5, Line 118-120:

The present study showed that the D-Ca in the range of 2.5 to 3.0 mEq/L did not impact the primary and secondary outcomes of the J-DAVID participants both the users and non-users of VDRA.

Revised, Page 5, Line 127-129:

The present study showed that the D-Ca in the range of 2.5 to 3.0 mEq/L did not have statistically significant impact on the primary and secondary outcomes of the J-DAVID participants in either the users or non-users of alfacalcidol.

Change #9 

Treatment with alfacalcidol was not based on the allocation group, but it was handled as a time-varying variable in this post hoc analysis, our careless mistake has been corrected. 

Original, Page 5, Line 128-130:

However, we noticed a numerically higher hazard for the composite of cardiovascular events associated with a higher D-Ca in patients who were allocated to the control group (without VDRA), not in patients who were allocated to the intervention group (using alfacalcidol), …

Revised, Page 6, Line 137-139:

However, we noticed a numerically higher hazard for the composite of cardiovascular events associated with a higher D-Ca in patients who were not treated with alfacalcidol, not in patients who were treated with alfacalcidol, …

Change #10

We have discussed on the possible influences of intravenous VDRAs and statistical power on the results in the revised manuscript. 

Original, Page 6, Line 132-137:

Although the exact mechanisms for these observations are unknown, the calcium load from dialysate may not have a detectable impact on clinical outcomes in the range of 2.5 to 3.0 mEq/L of D-Ca. 

According to the JSDT data in 2009, D-Ca was 2.5 to 3.0 mEq/L in more than 96% of the total number of reported dialysis patients (N=209,322) in Japan. Another observational study reported that the difference in D-Ca was not related to the survival similar finding to this study. (16)

Revised, page 6, Line 140-157: 

Although the exact mechanisms for these observations are unknown, the calcium load from dialysate may not have a detectable impact on clinical outcomes in the range of 2.5 to 3.0 mEq/L of D-Ca. According to the JSDT data in 2009, D-Ca was 2.5 to 3.0 mEq/L in more than 96% of the total number of reported dialysis patients (N=209,322) in Japan. Otherwise, the result might have been affected the fact that intravenous VDRAs other than oral alfacalcidol were allowed to be used if needed to follow the JSDT guideline, and that more patients in the control group received intravenous VDRAs than the intervention group of the original J-DAVID trial. 

This post hoc analysis was not able to show statistically significant association of D-Ca with composite cardiovascular events or all-cause mortality. The lack of significant associations may be attributable to the relatively low statistical power of the original J-DAVID trial. To address this issue, we took another outcome including both the primary composite cardiovascular outcome and the secondary all-cause mortality in this pos hoc analysis. However, the association of D-Ca and the combined outcome was again not significant. Similar to our results, another observational study derived from the Japanese Dialysis Outcomes and Practice Patterns Study reported that the difference in D-Ca was not related to the survival based on the results of 9,201 patients with a median follow-up of 2.03 years (16). Thus, the low statistical power does not fully explain the neutral results on the D-Ca in this range and clinical outcomes in this study. 

Change #11

Original, Page 8, Line 197:

Funding: This study was supported by the grant to TS from The Kidney Foundation, Japan.

Revised, Page 9, Line 217-218:

Funding: This study was supported by a grant to TS from The Kidney Foundation, Japan (no specific grant number was given). 

Change #12 

In the part of Competing interests, we have added the following statement.

Revised, Page 10, Line 245:

This does not alter our adherence to PLOS ONE policies on sharing data and materials. 

Change #13

Data sharing statement has been included in the revised manuscript.

Revised, Page 10, Line 246-247:

Data sharing: The data that support the findings of this study are available from the corresponding author of the original J-DAVID trial (TS) upon reasonable request.

Change #14 

Tables 1a, 1b, and 1c in the original manuscript has been combined into Table 1 in the revised version. The important contents in the original big tables have been edited to be more concise. The revised Table 1 has been included in the main body of the manuscript. 

Change #15 

Tables 2a and 2b in the original manuscript has been edited into Table 2 and Table 3 in the revised version. In the original version, the tables were created according to the outcome. In this revision, these tables were re-organized according to the exposure. The revised tables have been included in the main body of the manuscript. 

Change #16 

Our careless mistake (not mean but median) in the figure legend has been corrected.

Original, Page 13, Line 331-332:

Figure 1. Serial changes in the mean (25%, 75%) of serum levels of corrected Ca (1a), phosphate (1b), and intact PTH (1c) in D-Ca 2.5 and 3.0 mEq/L.

Revised, Page 18, Line 367-368:

Figure 1. Serial changes in the median (25th, 75th percentile) of serum levels of corrected Ca, phosphate, and intact PTH in D-Ca 2.5 and 3.0 mEq/L.

Change #17 

Because Figures 2 and 3 in the original version have been combined into new Figure 2 in this revision, Figure numbers and legends have been updated accordingly. 

Original, Page 13-14, Line 333-338:

Figure 2. Estimated hazards of all-cause mortality and non-fatal cardiovascular disease events by the use of alfacalcidol and the baseline dialysate Ca. There were no significant differences in the effects depending on the baseline dialysate Ca value. Treat: VDRA (+), control: VDRA (-)

Figure 3. Estimated hazards of all-cause mortality by the use of alfacalcidol and the baseline dialysate Ca. There were no significant differences in the effects depending on the baseline dialysate Ca value. Treat: VDRA (+), control: VDRA (-)

Revised, Page 18, Line 369-372:

Figure 2. Estimated hazards of composite cardiovascular events (primary outcome), all-cause mortality (secondary outcome), and the combined outcome (primary plus secondary) by the use of alfacalcidol and the baseline dialysate calcium concentration. Estimated hazard was plotted against the dialysate calcium concentration stratified by the use of alfacalcidol.

---

## [Decision Letter · Decision Letter 1]

4 Aug 2022

Dialysate Calcium, Alfacalcidol, and clinical Outcomes: A Post-Hoc Analysis of the J-DAVID Trial

PONE-D-22-04275R1

Dear Dr. Kunitoshi Iseki,

We’re pleased to inform you that your manuscript has been judged scientifically suitable for publication and will be formally accepted for publication once it meets all outstanding technical requirements.

Kind regards,

Larry Allan Weinrauch, MD

Academic Editor

PLOS ONE

Additional Editor Comments (optional):

Reviewers' comments:

Reviewer's Responses to Questions

**Comments to the Author**

1. If the authors have adequately addressed your comments raised in a previous round of review and you feel that this manuscript is now acceptable for publication, you may indicate that here to bypass the “Comments to the Author” section, enter your conflict of interest statement in the “Confidential to Editor” section, and submit your "Accept" recommendation.

Reviewer #3: All comments have been addressed

2. Is the manuscript technically sound, and do the data support the conclusions?

Reviewer #3: (No Response)

3. Has the statistical analysis been performed appropriately and rigorously? 

Reviewer #3: (No Response)

4. Have the authors made all data underlying the findings in their manuscript fully available?

Reviewer #3: (No Response)

5. Is the manuscript presented in an intelligible fashion and written in standard English?

Reviewer #3: (No Response)

6. Review Comments to the Author

Reviewer #3: (No Response)

7. PLOS authors have the option to publish the peer review history of their article (what does this mean?). If published, this will include your full peer review and any attached files.

Reviewer #3: No

---

## [Editor Report · Acceptance letter]

26 Aug 2022

PONE-D-22-04275R1 

Dialysate Calcium, Alfacalcidol, and clinical Outcomes: A Post-Hoc Analysis of the J-DAVID Trial 

Dear Dr. Iseki:

I'm pleased to inform you that your manuscript has been deemed suitable for publication in PLOS ONE. Congratulations! Your manuscript is now with our production department. 

Kind regards, 

on behalf of

Dr. Larry Allan Weinrauch 

Academic Editor

PLOS ONE